# Parameters to Predict the Outcome of Severe and Critical COVID-19 Patients when Admitted to the Hospital

**DOI:** 10.3390/jcm12041323

**Published:** 2023-02-07

**Authors:** Sonia del Carmen Chávez-Ocaña, Juan Carlos Bravata-Alcántara, Iliana Alejandra Cortés-Ortiz, Arturo Reyes-Sandoval, Jazmín García-Machorro, Norma Estela Herrera-Gonzalez

**Affiliations:** 1Genetics and Molecular Diagnosis Laboratory, Juárez de Mexico Hospital, Mexico City 07760, Mexico; 2Molecular Oncology Lab, Escuela Superior de Medicina, Instituto Politécnico Nacional, Plan de San Luis y Díaz Mirón s/n, Col. Casco de Santo Tomás, Ciudad de México 11340, Mexico; 3Bacteriology Laboratory, Juárez de Mexico Hospital, Mexico City 07760, Mexico; 4The Jenner Institute, University of Oxford, Old Road Campus Research Building, Roosevelt Drive, Oxford OX3 7DQ, UK; 5Instituto Politécnico Nacional, IPN, Av. Luis Enrique Erro s/n, Unidad Adolfo López Mateos, Mexico City 07738, Mexico; 6Laboratorio de Medicina de Conservación, Escuela Superior de Medicina, Instituto Politécnico Nacional, Mexico City 11340, Mexico

**Keywords:** COVID-19, severe, critical, progressing, death

## Abstract

Manifestations of COVID-19 are diverse and range from asymptomatic to severe, critical illness and death. Cases requiring hospital care (in severe and critical illnesses) are associated with comorbidities and hyperactivation of the immune system. Therefore, in this exploratory observational study, we analyzed which parameters are associated with mortality. We evaluated: demographic characteristics (age, sex and comorbidities), laboratory data (albumin, leukocytes, lymphocytes, platelets, ferritin), days of hospital stay, interleukins (IL-2, IL-6, IL-7, IL-10, IL-17) and sP-selectin in 40 Mexican patients admitted to medical emergencies with a confirmed diagnosis of COVID-19, a complete clinical record, and who signed the informed consent. Twenty severe (they required intermediate care with non-invasive ventilation) and twenty critically ill patients (they required mechanical ventilation) were classified, and these were subsequently compared with healthy and recovered subjects. A significant difference was found between the hospitalized groups in the parameters of age, ferritin, days of hospital stay and death with *p* values = 0.0145, *p* = 0.0441, *p* = 0.0001 and *p* = 0.0001, respectively. In the determination of cytokines and P-selectin, a significant difference was found between the following groups: recovered patients and healthy volunteers compared with hospitalized patients in severe and critical condition. Importantly, IL-7 remained elevated one year later in recovered patients. Taken together, these values determined at the time of hospital admission could be useful to monitor patients closely and evaluate in-hospital progress, hospital discharge, and out-of-hospital progress.

## 1. Introduction

The main role of the immune system is to recognize the presence of a foreign agent in the body and provoke an inflammatory response to promote its elimination. This action is orchestrated by cytokines, an extraordinary language through which immune cells coordinate and communicate with each other. There must be an extremely fine balance between the production of pro-inflammatory and anti-inflammatory cytokines, to protect the body from injury. The failure of the mechanisms that regulate the production of these cytokines can lead the immune system to overreact with a massive production of cytokines, causing an inflammatory reaction that can become systemic with terrible consequences for everything in the body [1]. 

Severe COVID-19 cases have been characterized by an immune response associated with a high production of pro-inflammatory cytokines. Also, numerous studies have shown that COVID-19 patients have increased levels of IL-1β, IL-2, IL-6, IL-10, TNFα, IFNγ, and that these cytokines correlate with the severity of the disease. When the production of these cytokines is extremely high it has been called “cytokine storm”. An inadequate regulation of the immune response to SARS-Cov2 virus has been proposed as a possible mechanism in the development of a cytokine storm during the infection [2].

In severe cases of COVID-19, the infection of lung cells can cause the recall of a rich inflammatory cell infiltration of neutrophils, macrophages, CD4+, CD8+T lymphocytes and massive production of cytokines, leading to bilateral pneumonia, acute respiratory distress syndrome (ARDS) and multi-organ damage. A significant proportion of hospitalized COVID-19 patients show an hyperinflammatory state associated with severe pneumonia and a high mortality rate [3]. Moreover, previous reports have also found widespread thrombosis and microangiopathy in the pulmonary vascular tissue of severe COVID-19 patients [4].

SARS-CoV-2 infects human cells by binding itself to the angiotensin-converting enzyme 2 (ACE-2) receptor, mainly expressed on respiratory epithelial cells, lungs, heart, kidney, intestine and other cells, as important as the endothelial cells [5]. Direct infection of endothelial cells, as well as the inflammatory environment, might result in an endothelial activation, P-selectin expression, increase of platelet recruitment to the site and finally, in the aggregation of platelet cells [6]. In fact, in the presence of severe infection or cytokine storm, platelet hyperreactivity may be the cause for cardiovascular adverse effects.

P-selectin, also known as CD62P, plays an important role in modulating interactions between blood cells and endothelial cells [7]. It is also constitutively present in alfa granules of platelets and in Weibel–Palade bodies in endothelial cells [7], and also found in human plasma where it is known as soluble P-selectin (sP-selectin).

Viral infections are known to be associated with coagulation disorders. For example, some coagulopathies were observed in SARS-CoV patients including thrombocytosis, disseminated intravascular coagulation and thromboembolism [8]. Several studies suggest a main role of platelet/endothelial interaction as part of the pathogenic mechanism of COVID-19. These interactions increase the leukocyte recruitment and the chemokine and cytokine release from the endothelial cells, which result in adhesion, activation and leukocyte trafficking through the endothelial wall. Thrombosis and coagulopathies have also been related to worse outcomes in COVID-19 patients.

The objective of our study was to investigate the demographic characteristics, laboratory parameters in conjunction with the interleukins IL-2, IL-6, IL-7, IL-10, IL-17, and sP-Selectin at the time of hospital admission and with this information be able to predict the possible severity of COVID-19 in each patient admitted to the emergency unit. We also proposed to evaluate the changes in cytokine and sP-selectin concentrations, one year after recovery.

## 2. Materials and Methods

Eighty adult patients were admitted to Hospital Juarez (Mexico) between April and June 2020, of which forty were included in this study. The inclusion criteria were an age of 18 years or older and a diagnosis of COVID-19 confirmed by means of a positive RT-qPCR for SARS-CoV-2 in at least one biological sample. They should not have history of drug treatment such as antibiotics and steroids. Additionally, they needed to have complete medical records, and a signed informed consent.

On the basis of the clinical, radiological (Reporting and Data System (CO-RADS) greater than 3) [9] and lab data, the severity of COVID-19 was considered severe when they required intermediate care with non-invasive ventilation, or critically ill when they were admitted to an intensive care unit for mechanical ventilation (according to the criteria reported by Shi et al. [10]).

On the day of admission, antecubital vein blood samples were collected into EDTA tubes for the measurement of the values of leukocytes, lymphocytes, platelets, albumin and ferritin. Plasma samples remained stored at −80 °C until determination of IL-2, IL-6, IL-7, IL-10, IL-17 and sP-selectin.

One year after diagnosis of SARS-CoV-2 infection, a plasma sample was taken to monitor interleukin and sP-selectin concentrations from ten severe patients who recovered. This was designated as the recovered group.

As a control group, ten samples from healthy volunteers, 18 years and older, with no history of SARS-CoV-2 infection (verified by negative rapid antibody test) were considered. At that time there were no vaccines against SARS-CoV-2. Then none of the volunteers had a history of SARS-CoV-2 vaccination. The group was designated as healthy.

The study was approved by the Ethics and Investigation Committee at the Juarez Hospital in Mexico City (No. HJM/ 001/2022) and was carried out in conformity with the 2018 revision of the Declaration of Helsinki and the code of Good Clinical Practice.

### 2.1. Data Collection

The characteristics of the population considered were age, sex, date of admission and discharge to quantify the days of hospital stay. Also considered were the history of comorbidities such as diabetes, hypertension, obesity in its different degrees, as well as the number of recovered patients and the number of deaths.

### 2.2. Interleukins and sP-Selectin Measurement

Plasma interleukins (IL-2, IL-6, IL-7, IL-10, IL-17) and sP-selectina were measured using Human ProcartaPlexMixandMatch 6-plex Assays panel and the Luminex 200 system (ThermoFisher Scientific, Waltham, MA, USA) for all patients according to the manufacturer’s instructions. Values were compared to plasma samples from ten healthy controls and ten recovered patients. Results are expressed as concentration in pg/mL.

### 2.3. Statistical Analysis

Statistical analyses were carried out using the GraphPad Prism software, version 6. Continuous variables were expressed as average ± standard error. Categorical variables were expressed as number (%) and compared by Mann–Whitney U test, between severe and critical groups. Boxplots were drawn to describe plasma interleukins and sP-selectin concentrations. Each dot represents an individual. The error bars denote the standard errors with a measurable concentration. A one-way un-matched ANOVA with a Tukey’s multiple comparisons test was used to determine the significance of the differences between the groups; the significance is denoted by asterisks as follows: * *p* < 0.05, ** *p* < 0.01, *** *p* < 0.001, **** *p* < 0.0001. 

## 3. Results

### 3.1. Demographics and Laboratory Findings of Patients Infected

Twenty patients were admitted to the Internal Medicine Service (diagnosed as severe) and 20 patients were admitted to the Intensive Care Service (diagnosed as critical). The characteristics of the population were age, sex, date of admission and discharge to quantify the days of hospital stay. Also considered were the history of comorbidities such as diabetes, hypertension, obesity in its different degrees, as well as the number of recovered patients and the number of deaths. The data of each patient is reported in Appendix A.

The characteristics of the population are summarized in Table 1. In the group of patients classified as severe, the minimum age was 18 years while the maximum age was 90 years, with an average of 49.50 ± 3.276. In the group of critical patients, the minimum and maximum age was 37 and 87 years, respectively, with an average of 61.00 ± 3.067 years. A significant difference was found in age between the two groups classified as severe and critical, with *p* < 0.05. Regarding sex, no significant difference was found between each group. In the health history, no significant difference was found between the groups.

Regarding the laboratory data, albumin was determined (reference values of 3.5–5.0 g/dL). In the severe and critical group, a minimum value of 1.4 and 1.8 g/dL, respectively, was found, with a maximum value of 4.8 and 4.6 g/dL, respectively. The average in the severe group was 3.595 ± 0.2089 g/dL and in the critical group was 3.265 ± 0.1658 g/dL, indicating no significant difference between the groups.

Leukocytes were considered with a reference value of 4500–11,000 cells/μL. In both groups, leukocytosis was found with an average of 11,695 ± 1073 and 13,315 ± 1416 in severe and critical, respectively. In the individual values, samples with a minimum value of 4800 cells/μL were found in the severe group and 3940 cells/μL in the critical group. No significant difference was found between the two groups.

Regarding the values of lymphocytes, in the severe and critical group, lymphocytopenia was found with minimum values of 170.0 cells/μL and 240.0 cells/μL, respectively. Even the low values are reflected in the average, and no significant difference was found between the two groups.

The average number of platelets in both groups was within normal values of 299,909 ± 26,825 cells/μL and 259,125 ± 19,314 cells/μL for severe and critical, considering the reference values of 150,000–400,000 cells/μL. However, when considering the individual values, thrombocytopenia of 3800 and 126,000 cells/μL was found in two patients belonging to the severe group. Likewise, in the serious group, two samples were found with high values of 543,000 and 601,500 cells/μL. In the critical group, two extreme values were found that fell outside the reference values, with a minimum of 128,000 cells/μL and a maximum of 425,000 cells/μL. No significant difference was found between the two groups.

In the determination of ferritin, a significant difference was found between the severe and critical groups with *p* = 0.0441, with an average of 574.5 ± 97.16 and 851.3 ± 90.76, respectively. Maximum values of 1500 μg/L were found (reference values of 24–336 μg/L).

In the quantification of days of hospital stay, a minimum value of 4 days was found for severely ill patients and a maximum of 36 days for critically ill patients, with an average of 10.10 ± 0.8611 and 19.80 ± 1.417 days for serious and critical, respectively. The significant difference was *p* = 0.0001. It is important to note that the coagulation alterations related to COVID-19, were seen early on in the pandemic, before the vaccines were available. When people started to get vaccinated, the number of serious cases decreased.

Unfortunately, 90% of the critical patients in the study died, as did 15% of the seriously ill patients. This outcome indicated a significant difference of *p* = 0.0001.

### 3.2. Interleukins and sP-Selectin

#### 3.2.1. IL-2 and IL-6 Concentration

The average concentration of IL-2 in the healthy group was 10.24 ± 6.90 pg/mL while in the recovered group it was 45.41 ± 6.44 pg/mL, with no significant difference between the two groups. The average concentration of the severe patients was 102.1 ± 9.27 pg/mL and that of the critical patients was 96.34 ± 9.45 pg/mL, with no significant difference between the two groups. However, when comparing the group of healthy vs. severe and healthy vs. critical, a significant difference was found with *p* < 0.0001. Likewise, a significant difference was found when comparing the group of recovered vs. severe and recovered vs. critical with *p* < 0.01 (see Figure 1a).

In the average determination of IL-6 in healthy and recovered patients (6.47 ± 5.85 pg/mL and 7.12 ± 4.14 pg/mL, respectively), there was no significant difference. However, when comparing the results of healthy vs. severe and healthy vs. critical, a significant difference was found with a value of *p* < 0.001 and *p* < 0.0001, respectively. In the comparison of recovered vs. severe and recovered vs. critical, the significant difference was *p* < 0.001 and *p* < 0.0001, respectively. No difference was found between severe and critical, with an average value of 129.4 ± 18.03 pg/mL and 133.0 ± 18.78 pg/mL, respectively (see Figure 1b).

#### 3.2.2. IL-7 and IL-10 Concentration

In the case of IL-7, a significant difference was found when comparing the healthy (average 3.38 ± 3.38) group with the recovered, severe and critical groups (with a value of *p* < 0.05, *p* < 0.0001 and *p* < 0.0001 respectively). The average value found was: recovered 28.50 ± 7.39 pg/mL, severe 47.52 ± 4.03 pg/mL and critical 46.29 ± 4.89 pg/mL; no significant difference was found between them (see Figure 2a).

In the determination of IL-10, no significant difference was found between the groups of healthy vs. recovered, with an average of 0.44 ± 0.44 pg/mL and 3.96 ± 0.82 pg/mL (see Figure 2b). However, when comparing healthy vs. severe, healthy vs. critical, recovered vs. severe, recovered vs. critical, a significant difference was found, with *p* < 0.0001. When comparing severe vs critical, there was no significant difference (with an average of 18.50 ± 1.96 and 15.27 ± 1.12 pg/mL, respectively).

#### 3.2.3. IL-17 and sP-Selectin Concentration

IL-17 was found to be elevated in samples from severe and critical patients, with an average of 312.8 ± 29.93 pg/mL and 297.0 ± 30.62 pg/mL, with no significant difference between both groups. However, when comparing healthy vs. severe, healthy vs. critical, recovered vs. severe, recovered vs. critical, the significant difference was *p* < 0.0001. The average value of IL-17 in healthy and recovered patients was 11.05 ± 7.36 pg/mL and 35.93 ± 6.55 pg/mL, respectively, with no significant difference between them (see Figure 3a).

The average value of sP-selectin in the samples from healthy patients was 1446 ± 565.6 pg/mL, with no significant difference between healthy vs. recovered patients. However, when comparing healthy vs. severe, healthy vs. critical, a significant difference of *p* < 0.0001 and *p* < 0.001, respectively, was found. When comparing recovered vs. severe, recovered vs. critical, a significant difference of *p* < 0.0001 and *p* < 0.001, respectively, was found. The average value of severe and critical was 334,233 ± 47,117 pg/mL and 254,017 ± 32,578 pg/mL, respectively, without finding a significant difference between the two (Figure 3b).

## 4. Discussion

In this study we present the analysis of 40 patients confirmed for SARS-CoV-2 who required hospitalization at Hospital Juarez Mexico. Twenty patients were classified as severe and 20 as critical according to the criteria reported by Shi et al. [10]. The samples were collected from April to June 2020. In addition to comorbidities, laboratory values and interleukins were investigated. Therefore, our work covers other aspects that have not been reported and includes patient samples before the application of vaccines against SARS-CoV-2.

Regarding age of patients, a significant difference was found (*p* = 0.0145), with the patients classified as critical being older (average age 61.00 ± 3.067 years) than the severe ones, who had an average age of 49.50 ± 3.276 years; similar data have been reported [11,12], that COVID-19 is of greater risk for elderly patients. In terms of sex characteristics, we did not find a significant difference in severe or critical illness, unlike other studies that have reported that male individuals are more likely to be hospitalized in the ICU [12]. Another study showed that aged and male individuals are more likely to be hospitalized as intensive care unit patients [13]. This difference may be due to the type of comorbidities that affect the Mexican population.

Diabetes, systemic hypertension, and obesity are the most common comorbidities in Mexico [14], and for this reason they were investigated, and indeed most patients had at least one comorbidity (except for two of the 40 patients who did not have any comorbidity). Diabetes in patients with COVID-19 is associated with a twofold increase in mortality and severity of COVID-19, compared to non-diabetics [15,16]. This coincides with the data found, since in the severe group, 11 patients were diabetic and 3 died; and in the critical group, 8 were diabetic and 7 died.

Hypertension has been associated with mortality from COVID-19 [17] and indeed, in this investigation we found 10 hypertensive patients in total, of which 5 were severe (none died) and 5 critical (4 died). The low prevalence of hypertension in this population may be due to under-registration as the patient may be unaware of the diagnosis. Despite there being levels equal to or greater than 140/90 mmHg, the diagnosis of hypertension should be based on taking several measurements made on separate occasions [18,19]. At the time the patients were admitted to the hospital, we did not diagnose hypertension since there are some factors, such as pain, that can increase blood pressure without the patient being hypertensive.

Since the 2009 pandemic due to influenza A H1N1, obesity has drawn attention since it was found that obese people were more susceptible to infection by respiratory viruses, with greater severity of the disease [20,21]. In this pandemic due to the SARS-CoV-2 virus, obesity has also been reported in patients with severe disease [22]. In the US, it was reported that obese patients with SARS-CoV-2 are more likely to be admitted to the ICU than those with a BMI < 30 kg/m^2^ [23]. The data coincides in other places such as China [24], Latin America [25], etc. In the results of this work, obesity was found in its different degrees without finding a significant difference between severe and critical. Interestingly, in terms of mortality, none of the total number of obese patients (three obese patients) who became severely ill died, unlike the 6 critically obese patients, who unfortunately all died.

Additionally, some laboratory data were determined for the patients, such as albumin, lymphocytes, platelets, and ferritin; these factors are of interest for their possible correlation with the severity of COVID-19.

Albumin is responsible for 80% of the oncotic pressure in the vessels. This is necessary to maintain enough water within the circulatory system and for the maintenance of sufficient blood pressure, as well as for a sufficient blood supply for vital organs such as the brain, lungs, heart and kidneys. The liver reacts to a decrease in oncotic pressure with an increase in albumin synthesis. Acute diseases cause the activation of defense mechanisms (known as acute phase reaction) that can lead to increased fibrinolysis and an increase in the plasma level of fibrinogen degradation products, mainly fibrin and D-dimer [26]. Albumin measurement has been used as a prognostic marker that may show a predisposition to venous thromboembolism [27] due to increased fibrinolysis that may occur at peripheral ischemic sites, where clotting proteins may be part of the interim clot (e.g., in the lungs).

Bannaga et al. [28] reported on measurement of serum D-dimer level, and this test was introduced into the routine laboratory process of hospitalized COVID-19 patients early on as part of patient characteristics in early reports. Hariyanto et al. [29] published the results of their meta-analysis on inflammatory and hematological markers as predictors of severe outcomes in COVID-19 patients. Interestingly, a serum albumin level below 38.85 g/L was also a negative prognostic marker. In their discussion, the authors mention its possibility as a marker of rehospitalization and embolism after discharge from hospital and in association with other fibrinolytic markers.

In our study we observed that albumin levels are lower in the group of critical subjects with respect to the group of severe subjects, although the difference is not statistically significant; this may be because the levels were taken at the beginning of their hospitalization and as described by Violi et al. [27], serum albumin levels decrease sharply over several days of hospitalization.

Additionally, in coagulation problems, alterations in the number of platelets have been reported, including findings that the platelet count is much lower in the deceased compared to the survivors [30,31]. However, in this study, the average of the patients presented platelet values within normal ranges. In the group of severely ill patients, two patients with thrombocytopenia (38,000 and 126,000 cells/μL) and two with thrombocytosis (543,000 and 601,500 cells/μL) were found. Fortunately, none died. In contrast, in the group of critical patients, one patient presented thrombocytopenia (128,000 cells/μL) and another had thrombocytosis (425,000 cells/μL) and both died. With the above, we can see that not all studies agree that platelets can be a predictor of mortality from COVID-19 [32].

Regarding leukocyte quantification, leukocytosis was found in 55% of patients classified as severe, and in 60% of critical patients, with an average value of 11,695 ± 1073 and 13,315 ± 1416 cells/μL, respectively; no significant difference was found between groups. However, there are reports where the white blood cell count is higher in patients who do not survive than in those who survive [33,34,35]. Therefore, survival was compared and it was found that of the 11 severe patients with leukocytosis, 2 died; in contrast, of the 12 critical patients with leukocytosis, 11 died. Although there is no significant difference in the average between the two groups, leukocytosis in critically ill patients is higher than in severely ill patients and coincides with what has been reported.

Various studies have reported a statistically significant reduction in total lymphocytes in patients with severe/critical disease due to COVID-19 compared to mild/moderate disease [3,36,37]. In this study, a reduction in lymphocytes was found in both the severe and critical groups with an average of 995.5 ± 130.3 and 831.8 ± 108.5 cells/μL, respectively, with no significant difference between the two groups.

Ferritin is the main iron storage protein; its serum level reflects the normal level of iron. Various studies found an association between serum ferritin level and severity of COVID-19 disease, including mortality [38,39]. Indeed, in this study it was found to be higher in critical patients (851.3 ± 90.76 μg/L) than in severe patients (574.5 ± 97.16 μg/L), with a significant difference between both groups of *p* = 0.0441.

The days of hospital stay are variable, since they depend on the admission criteria, hospital discharge criteria, local capacity and the pressure on the health system. In this study, severely ill patients spent an average of 10.10 ± 0.8611 days hospitalized in ICU, whereas critical patients spent an average of 19.80 ± 1.417 days in ICU. These data are similar to those reported in Lombardy, Italy. The authors reported the median length of ICU stay was 12 (95% CI, 12–13; IQR, 6–21) days; however, the median of stay of patients who died in the ICU was 10 days [40]. In another study conducted in China in patients who became ill from January 16, 2020 to March 4, 2020, the median length of stay for all confirmed inpatients was 19 days [41].

Unfortunately, the number of deaths found in this work is very high in critical patients (90%) compared to severe patients (15%). In a study in the city of Pune, India, from March to December 2020, gross hospital mortality from COVID-19 was 8.17%, of which 107 (65.2%) patients died from severe COVID-19 and 57 (34.75%) died from critical illness [42]. In the New York City area from March to April 2020, 21% of COVID-19 deaths of all hospitalized patients were reported [43]. However, mortality data should not be lightly compared, since they depend on various factors such as comorbidities, hospital admission, even if the report is for all patients, only hospitalized or hospitalized in the ICU.

The immunological profile of severely ill patients with COVID-19 has been described in multiple trials and suggests a hyperactivation of the humoral immune pathway, including several interleukins [44]. While these molecules are important for eliminating the virus, they can also cause tissue damage when released to inappropriate levels [45]. 

In this work we found elevated proinflammatory cytokines IL-6, IL-10, IL-2 and IL-17, and importantly we noted that 12 of 20 COVID-19 severely ill patients and 13 of 20 critically ill patients had lymphopenia (lymphocyte values below 1000/ mL), which coincides with the observations of other authors [46]. 

Severe and critical groups were classified according to ventilatory and severity parameters described [10]. In interleukins values, we did not find a statistically significant difference between the groups, possibly because both groups required hospital care having this very homogeneous criterion and the samples were taken upon admission, so it could take days for the interleukins’ response to be reflected in the hyperactivity of the system. This is in contrast to findings reported by Huang et al. [47] where ICU patients had higher plasma levels of IL-2, IL-7, IL-10, GSCF, IP10, MCP1, MIP1A, and TNFα. 

The significant increase in IL-6 has been described as an important biomarker for severe or critical patients [48,49,50]. This result coincides with our results with respect to hospitalized patients. However, we found no difference in hospitalized patients classified as severe or critical. We believe that the difference found is due to the manner in which patients are classified. In a study in Mexican patients [49], the authors found a difference in the concentration of IL-6 between critical and non-critical patients. However, the authors considered both the moderate and the severe as non-critical, whereas critical definition includes gravity signs plus the need for mechanical ventilation, and/or the development of shock or multiple organ dysfunction syndrome. In our work, severe patients do not include moderately ill patients; the classification was made according to reporting by Shi, Y. et al. [10].

In another article [50], the authors reported a difference in the concentration of IL-6 in COVID-19 patients admitted to the ICU with hypoxemic respiratory failure compared to reference values. Thus, the comparison is with the reference values, without classifying hospitalized patients between severe and critical. Therefore, our results support the reported results with respect to hospitalized patients. However, we found no difference in the concentration of IL-6 between hospitalized patients classified as critical and severe.

However, the value of this study is to compare the values one year later in recovered patients and in a group of volunteers with no history of COVID-19 or vaccination.

In recovered individuals, the IL-2 concentration was slightly higher than in the group of healthy people (no significant difference found). This might be due to the persistence of inflammation in post-COVID conditions, which suggests that it would be convenient to follow these subjects for a while after their recovery.

In the determinations of IL-6, IL-10 and IL-17, values similar to the healthy group are reached, except in IL-7. IL-7 has also been observed as a presence value for cardiological damage post-COVID [51]. If there is a statistically significant difference between healthy and recovered subjects, we believe it may be possibly a sequel damage of the myocardium in the recovered subjects, which suggests a target for future research.

Determining sP-selectin was also of interest, since it is stored and expressed by endothelial cells [7]. Elevated levels of sP-selectin may also reflect platelet activation, as *p*-selectin is proteolytically detached from the plasma membrane in vivo shortly after activation [52]. Plasma levels of sP-selectin have also been considered a useful biomarker in cardiovascular disease [52]. Thus, the increase in sP-selectin may also reflect the activation and damage of endothelial cells. Pathological levels of sP-selectin consistently promote leukocytes to adhere to the endothermium through activation of Mac-1 leukocyte integrin [53]. Circulating sP-selectin is also thought to trigger signaling in leukocytes that has a direct contribution to inflammation and thrombosis [52].

Considering the optimal threshold value 35,506 pg/mL of sP-selectin determined as predictor of mortality in heart failure by Kanagala et al. [54], the value is high in severe and critical patients (334,233 ± 47,117 pg/mL and 254,017 ± 32,578 pg/mL, respectively). On the other hand, Watany et al. [55] reported sP-selectin in COVID-19 patients with thrombosis and patients without thrombosis at 6.31 ng/mL and 2.43 ng/mL, respectively. This indicates that patients classified as severe and critical in this study are at risk of thrombosis, while recovered patients and healthy volunteers are without risk (with optimal values).

The set of characteristics evaluated is illustrated in Figure 4. In severe patients the increase in ferritin, the greater number of days of hospital stay, and older age indicate the evolution to critically ill.

## 5. Conclusions

In patients admitted to medical emergencies with a confirmed diagnosis of COVID-19, it is important to perform a joint evaluation of demographic characteristics, clinical parameters, interleukins (IL-2, IL-6, IL-7, IL-10, IL-17) and sP-selectin. Together, these values determined in hospital admission could be useful to closely monitor patients, evaluate in-hospital progress, hospital discharge, and out-of-hospital progress. In the medium term, early medical intervention can be considered in critically ill patients, which could be useful if these tests were routinely implemented for follow-up in critically ill patients.

## Figures and Tables

**Figure 1 jcm-12-01323-f001:**
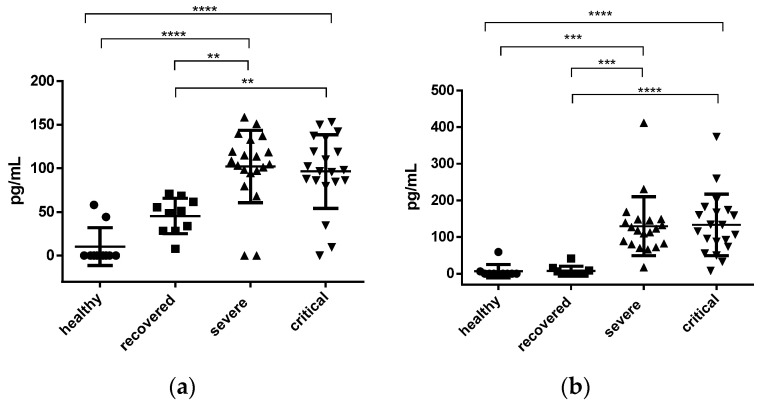
Plasma interleukin IL-2 and IL-6 concentration. (**a**) IL-2 concentration; (**b**) IL-6 concentration. The results are expressed as concentration pg/mL. Each dot represents a plasma sample, the circles represent the samples of the healthy group, the squares the recovered ones, the triangles those classified as severe and the inverted triangles the samples of critical patients. The error bars denote the standard errors with a measurable concentration. A one-way un-matched ANOVA with a Tukey’s multiple comparisons test was used to determine the significance of the differences between the groups; the significance is denoted by asterisks as follows: ** *p* < 0.01, *** *p* < 0.001, **** *p* < 0.0001.

**Figure 2 jcm-12-01323-f002:**
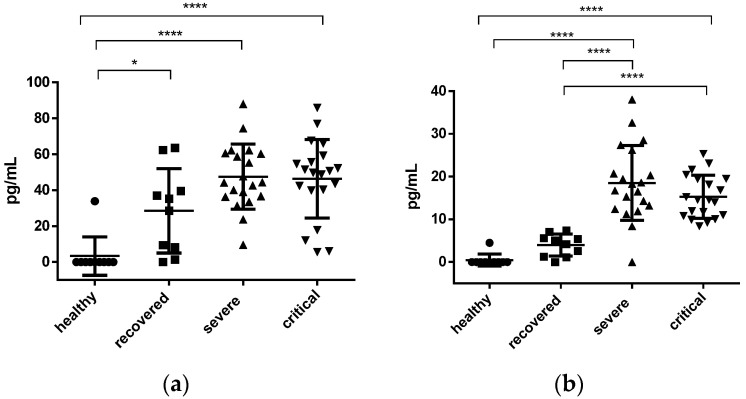
Plasma interleukin IL-7 and IL-10 concentration. (**a**) IL-7 concentration in pg/mL; (**b**) IL-10 concentration pg/mL. Each dot represents a plasma sample, the circles represent the samples of the healthy group, the squares the recovered ones, the triangles those classified as severe and the inverted triangles the samples of critical patients. The error bars denote the standard errors with a measurable concentration. A one-way un-matched ANOVA with a Tukey’s multiple comparisons test was used to determine the significance of the differences between the groups; the significance is denoted by asterisks as follows: * *p* < 0.05, **** *p* < 0.0001.

**Figure 3 jcm-12-01323-f003:**
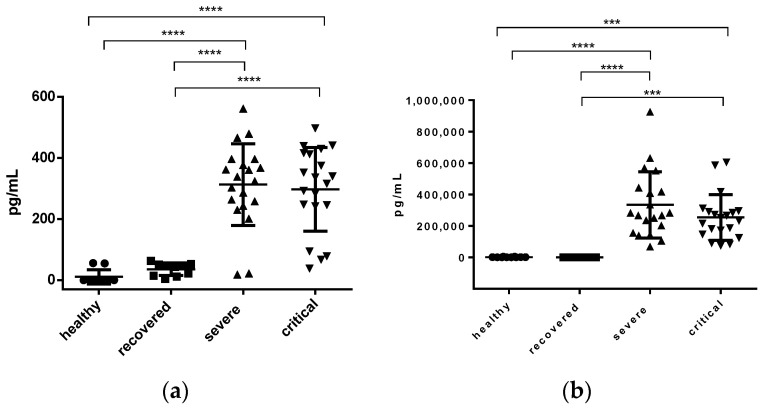
Determination of soluble molecules in patient plasma. (**a**) IL-17 concentration in pg/mL. (**b**) sP-selectina concentration in pg/mL. The significance is denoted by asterisks as follows: *** *p* < 0.001, **** *p* < 0.0001.

**Figure 4 jcm-12-01323-f004:**
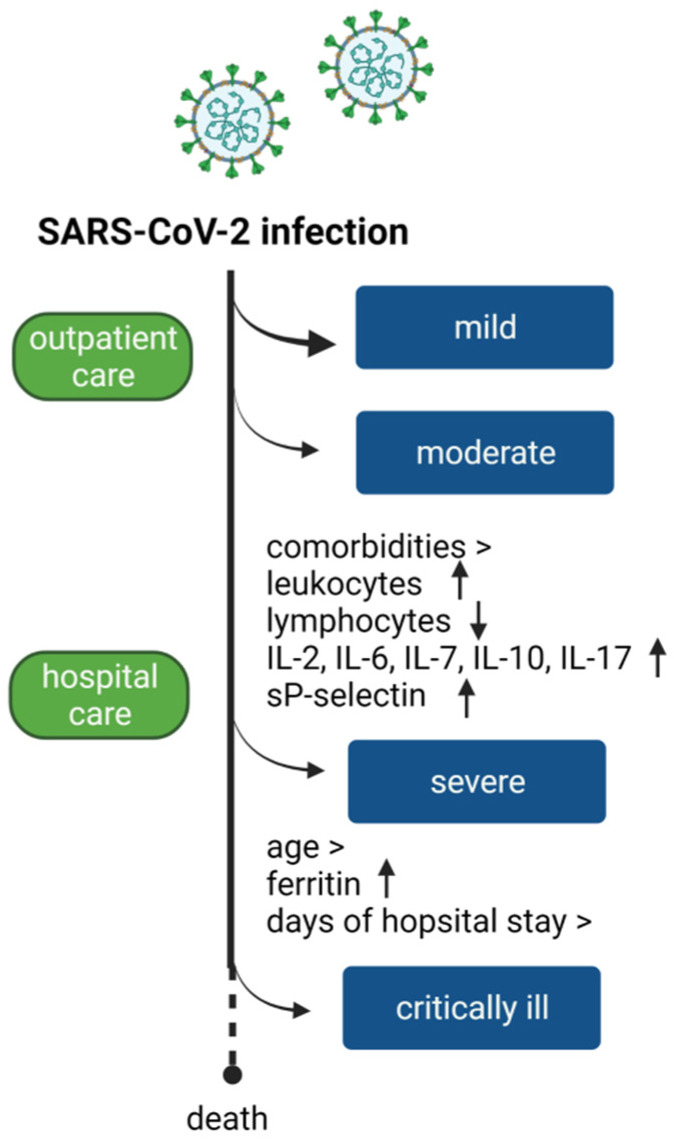
SARS-CoV-2 infection is variable from mild to moderate, requiring outpatient care. However, severe and critical illness requires hospital care. According to our results, comorbidities, leukocytosis, lymphopenia, increased interleukins, sP-selectin indicate severity. Additionally, at older ages, an increase in ferritin values and an increase in days of hospital stay indicate an evolution to critical illness.

**Table 1 jcm-12-01323-t001:** Demographics and laboratory findings of patients infected with SARS-CoV-2 on admission to hospital.

Characteristics	All Patients (n = 40)	Severe (n = 20)	Critical (n = 20)	*p* Value
Age, years	55.25 ^a^ ± 2.399 ^b^	49.50 ± 3.276	61.00 ± 3.067	0.0145 *
Sex	Men 22 ^c^ (55%) ^d^	8 (40%)	14 (70%)	0.1110
	Women 18 (45%)	12 (60%)	6 (30%)	
Any comorbidity	Diabetes 19 (47.5%)	11 (55%)	8 (40%)	0.5273
	Hypertension 10 (25%)	5 (25%)	5 (25%)	>0.999
	Obesity I 1 (2.5%)	1 (5%)	0	>0.999
	Obesity II 4 (10%)	2 (10%)	2 (10%)	>0.999
	Obesity III 3 (7.5%)	0	3 (15%)	0.2308
	Obesity IV 1 (2.5%)	0	1 (5%)	>0.999
Laboratory data	Albumin ^e^ 3.430 ± 0.1343	3.595 ± 0.2089	3.265 ± 0.1658	0.2236
	Leukocytes ^f^ 12,505 ± 886.4	11,695 ± 1073	13,315 ± 1416	0.3677
	Lymphocytes ^g^ 913.6 ± 84.70	995.5 ± 130.3	831.8 ± 108.5	0.3404
	Platelets ^h^ 279,517 ± 16,638	299,909 ± 26,825	259,125 ± 19,314	0.2248
	Ferritin ^i^ 712.9 ± 69.26	574.5 ± 97.16	851.3 ± 90.76	0.0441 *
Days of hospital stay	14.95 ± 1.128	10.10 ± 0.8611	19.80 ± 1.417	0.0001 ****
Death	21 (52.5%)	3 (15%)	18 (90%)	0.0001 ****

The characteristics of age, laboratory data and days of hospital stay are expressed as: ^a^ Average, ^b^ Standard Error. The characteristics of sex, comorbidity and death are expressed as: ^c^ number of patients, ^d^ percentage. Reference values: ^e^ 3.5–5.0 g/dL, ^f^ 4,500–11,000 cells/μL, ^g^ 1000–4800 cells/μL, ^h^ 150,000–400,000 cells/μL, ^i^ 24–336 μg/L. * *p* < 0.05, **** *p* < 0.0001.

## Data Availability

Not applicable.

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
