# Peer review of "Parameters to Predict the Outcome of Severe and Critical COVID-19 Patients when Admitted to the Hospital"

_jcm, 2023, doi:10.3390/jcm12041323_

Round 1

Reviewer 1 Report

Dear Editor,

I have read with great interest the manuscript entitled “Parameters to predict the outcome of severe and critical COVID-19 patients when admitted to the hospital” submitted for consideration on your Journal.

Authors deal with an interesting topic highly discussed in the recent literature.

Though some data could be of interest, most comprehensive papers have been published in this field and some doubts remain about the novelty of this research.

Along with these observations some aspects need to be more deeply reviewed from the investigators (i.e. How the Authors explain the low prevalence of systemic hypertension in their population – particularly  in critical COVID 19 - ; how the Authors explain the absence of significant alteration in IL-6 which has been described as important biomarker for severe or critical patients).

Also, the difference from severe and critical COVID-19 should follow WHO statements while the Authors based this differentiation according the ward or the necessity for ICU.

Data presentation is absolutely insufficient.  Other important biochemical data are not reported .

The same for clinical data (LUS Score, CHUNG score, P/F , etc)

It Is not clear also to me the study design . Did the Authos have repeated the ILs sample after one year. “Importantly, IL-7 remained elevated one year later in recovered patients” . Where are in results these data?

Finally, many typos are present also in the figure.

Reviewer 2 Report

Congratulation on your work. Please make the tĺetters of groups in the graph the same. I think that 40 patients are not too much, maybe you can make the experimental group bigger.

Reviewer 3 Report

Chávez-Ocaña et al. conducted an explorative study titled “Parameters to predict the outcome of severe and critical COVID-19 patients when admitted to the hospital”. The authors present an interesting manuscript about the association between some selected biomarkers and outcomes of 40 hospitalized Covid-19 patients.

Here are some of my comments:

1.     Abstract: Unfortunately, the abstract is quite difficult to understand. How were the 40 patients selected. 50% of the patients were critically ill. I assume this was not a random sample. Instead of “naked” p values, please provide the actual numbers to allow a better understanding of the observed differences. The conclusion that this may help to prevent progression to critical illness is an overstatement which has not been subject of this research. 

1.     The introduction is well written and stimulates the reader (a rare finding). However, in line 86-97 you already present/discuss your results. This is quite unconventional. I suggest removing these data and rather precisely define the aims of your work.

2.     Line 99: How where the patients selected? Have you performed a sample size calculation? I assume it is no coincidence that you have exactly 20 severe and 20 critical patients?

3.     Line 101: I assume the IC was age of 18 years or older? Because the youngest patient was not older than 18 years of age…

4.     How did you select healthy controls? How did you select recovered patients (how long was the time from infection to inclusion). Please provide more details.

5.     I think - in the context of your study - sex (instead of gender) should be the preferred term.

6.     Also in Line 198ff: Please provide actual numbers. P values are not very informative.

7.     You performed an explorative analysis about which parameters are associated with mortality in a univariate relationship. This is possible, but not the desired approach for prediction studies. AUROC analyses and multivariate regression analyses would be preferred to assess the predictive potential of the investigated markers. I think you can only omit these additional analyses if you cleary emphasize the explorative nature of your work and if you weaken your conclusion about the predictive value of the investigated biomarkers.

Round 2

Reviewer 3 Report

Thanks for responding to my comments. There are two more things that you have not adressed: 

1) Line 108 still says "older than 18 years". However, subject was 18 years old. Please change to "18 years or older"

2) I asked you for actual numbers in the discussion and Line 196 to 199. You commented that you added them but you did not. If you talk about significant difference, please provide the measure of central tendency and dispersion for both groups! (eg Group A had a mean of X ± Y, and Group B had a mean of W ± Z  (p = 0.00X)) 
